# Comparative Evaluation of Organic Acid Pretreatment of Eucalyptus for Kraft Dissolving Pulp Production

**DOI:** 10.3390/ma13020361

**Published:** 2020-01-12

**Authors:** Yuanhang Chen, Zhenyun Yan, Long Liang, Miao Ran, Ting Wu, Baobin Wang, Xiuxiu Zou, Mengke Zhao, Guigan Fang, Kuizhong Shen

**Affiliations:** 1Institute of Chemical Industry of Forest Products, Chinese Academy of Forestry, Key Lab of Biomass Engineering and Material Jiangsu Province, National Engineering Lab. for Biomass Chemical Utilization, Key and Open Lab of Forest Chemical Engineering, SAF, Nanjing 210042, China; chentuo26yh@163.com (Y.C.); chenhang0626@gmail.com (Z.Y.); lianglong@icifp.cn (L.L.); fmfyhg34169@chacuo.net (M.R.); wuting@icifp.cn (T.W.); pemshb98461@chacuo.net (X.Z.); iscpbg67531@chacuo.net (M.Z.); 2College of Light Industry Science and Engineering, Nanjing Forestry University, Nanjing 210037, China; jtvibz45217@chacuo.net

**Keywords:** eucalyptus pretreatment, organic acid, dissolving pulp

## Abstract

Pretreatment is an essential process for the extensive utilization of lignocellulose materials. The effect of four common organic acid pretreatments for Kraft dissolving pulp production was comparatively investigated. It was found that under acidic conditions, hemicellulose can be effectively removed and more reducing sugars can be recovered. During acetic acid pretreatment, lignin that was dissolved in acetic acid could form a lignin-related film which would alleviate cellulose hydrolysis, while other organic acids caused severe cellulose degradation. Scanning electron microscopy (SEM), Fourier transform infrared spectroscopy (FTIR) and X-ray diffractometry (XRD) were used to characterize the pretreated chips in the process. Lignin droplets were attached to the surface of the treated wood chips according to the SEM results. The FTIR spectrum showed that the lignin peak signal becomes stronger, and the hemicellulose peak signal becomes weaker with acid pretreatment. The XRD spectrum demonstrated that the crystallinity index of the wood chips increased. The acetic acid pretreatment process-assisted Kraft process achieved higher yield (31.66%) and higher α-cellulose (98.28%) than any other organic acid pretreatment. Furthermore, extensive utilization of biomass was evaluated with the acetic acid pretreatment-assisted Kraft process. 43.8% polysaccharide (12.14% reducing sugar and 31.66% dissolving pulp) and 22.24% lignin (0.29% acetic acid lignin and 21.95% sulfate lignin) were recovered during the process. Biomass utilization could reach 66.04%. Acetic acid pretreatment is a promising process for extensive biomass utilization.

## 1. Introduction

Given the global economic and environmental issues associated with the widespread use of petrochemicals, research on the utilization of biomass is increasing. Lignocellulose materials are the most abundant, renewable, biological resources on the planet, and they are suitable raw materials for sustainable human development [1,2,3]. Biorefineries fulfill the extensive utilization of biomass, and therefore provide additional economic and environmental benefits.

The prehydrolysis kraft process is a well acknowledged process for dissolving pulp production. Pretreatment is an indispensable step for dissolving pulp production. This process can effectively remove hemicellulose and disrupt the structure of the biomass. 

Various pretreatment processes were investigated. Generally, they are divided into four categories, including physical, chemical, physicochemical and biological methods [4]. However, physical pretreatments are not widely used, given the limited effectiveness [5,6]. In recent studies, microwave and ultrasound are also used in the pretreatment process. Karunanithy et al. treated switchgrass and big bluestem with extrusion followed by microwave irradiation. Compared to raw materials, it was found that the total sugar recovery was increased by at least 1.14 times through enzymatic hydrolysis, while the extra energy input introduced by microwave irradiation might not be economical [7]. Physicochemical treatments, such as steam-explosion pretreatment, are carried by high temperature and pressure, and can increase the specific surface area of the fiber. However, safety and cost issues might limit its practical application [8,9]. Biological pretreatment is usually carried out under milder conditions, but extended time (from days to weeks) is required to get the desired impact [10]. Wan et al. reported that Ceriporiopsis subvermispora could remove lignin from hardwood and retain cellulose well, but a long time (18 days) is needed during the pretreatment [11]. In the pulp production industry, hot water pretreatment and dilute acid pretreatment are the most widely used pretreatment methods. Hot pretreatment is an attractive method for treating lignocellulose biomass because of its environmental friendliness and low operating costs [12]. However, only a limited amount of hemicellulose can be removed, which is inferior for the hemicellulose utilization, and the cellulose will be seriously degraded with increased pretreatment intensity [13]. Similarly, dilute sulfuric acid is widely used to pretreat various broad-leaved wood and agricultural waste, because it is effective and economical to remove hemicellulose. However, the significant disadvantages of this method includes the production of inhibitory compounds and the corrosion of the reaction vessel [14,15,16].

Organic acids are increasingly used in biomass pretreatment, because of their efficiency in hydrolyzing lignocellulose biomass with less degradation products and more oligomeric sugars. Lin et al. compared different acid pretreatment methods (with oxalic acid, maleic acid, citric acid and sulfuric acid), and found that organic acid pretreatment makes it easier to recover hemicellulose from the waste liquor of dissolving pulp [17]. Kwon et al. pretreated the macro-algae Gracilaria verrucosa with citric acid and enzymatically hydrolyzed it to obtain a yield of 57.8% reducing sugar [18]. Lu et al. analyzed a maleic acid-catalyzed corn stover, and obtained a kinetic model of hemicellulose hydrolysis to obtain a xylose yield of 96% of the theoretical value [19]. This means that maleic acid is effective at removing xylan, which is necessary to prepare high-quality dissolving pulp, and is also efficient for recovering hemicellulose. It is feasible to produce dissolving pulp from organic acid due to the effective removal of hemicellulose. Through the selective catalysis of oxalic acid, Stein et al. achieved an efficient separation of wood fibers in a two-phase system consisting of water and 2-MTHF [20]. When Jahan et al. [21] treated the straw with a mixture of acetic acid/formic acid/water during the biomass refining process, the final cellulose performance reached 93.6% α-cellulose, 5.1% pentosan and 81.8% whiteness. Though there are many studies on the pretreatment of organic acids, a comparative investigation of the effect of organic acid pretreatment for the production of Kraft dissolving pulp is still limited.

In this study, the pretreatment effect of organic acids on wood fibers was compared with conventional hot water pretreatment and sulfuric acids pretreatment. Then, the effects of the pretreatment process were further evaluated by comparing the relative dissolving pulp properties. Finally, the mass balance of the acetic acid pretreatment process was evaluated for the extensive utilization of Eucalyptus in the Kraft dissolving pulp process.

## 2. Materials and Methods

### 2.1. Materials

Eucalyptus chips were obtained from the Hunan Province YuNong Paper Co, Ltd. (Luzhou, China), organic acid from the Yonghua fine chemicals Co, Ltd., (Suzhou, China), and all chemicals were of analytical grade.

### 2.2. Pretreatment Experiments

The wood powder (20–80 mesh) was treated with five different acids (pH = 2 or 3) at different temperatures (150 °C, 160 °C or 170 °C) for 2 h. The solid–liquid ratio was kept at 1:7.

The 2–4 cm eucalyptus chips were sealed to balance moisture before experiment. The deionized water and five different dilute acids were added into the pressure-resistant autoclave containing eucalyptus. The experiments was carried out at the pH value of 2 and at 160 °C for 2 h. The solid–liquid ratio was 1:7. 

### 2.3. Cooking

After the pretreatment, the eucalyptus chips were filled into the reaction kettle for cooking with 21% alkali and 17% sulfidity. The temperature was maintained at 130 °C for 0.5 h, and then heated to 160 °C for 2 h. After the reaction, the pulp was washed to neutral, and then they were sealed and stored in a black plastic bag for testing and bleaching.

### 2.4. Oxygen Delignification and ECF Bleaching

The original pulps were torn into small pieces to balance moisture. The bleaching process followed a typical ODEP process. The O (oxygen delignification) was carried out in a 500 mL high pressure reaction kettle. After the reactions were completed, the pulps were washed to neutral and squeezed for Elemental-Chlorine-Free bleaching. The section was conducted by DEP three-stage bleaching, in which D was chlorine dioxide treatment, E was an alkaline extraction and P was the hydrogen peroxide bleaching. More specifically, the pulps after oxygen delignification were shredded, then the liquid was fully mixed in the sealed bag, and the pulps were kneaded once every 10–15 min to ensure a uniform reaction. The specific process parameters are shown in Table 1. Lastly, the pulp was washed thoroughly until it was neutral (pH = 7.0), then we squeezed out the moisture, and put the pulp into a sealed bag to balance the moisture for testing.

### 2.5. Analysis Method

#### 2.5.1. Analysis of Pretreatment Components

After pretreatment, solids analysis was referenced with NREL [22]. The wood chips were ground to a size of 20–80 mesh. After being hydrolyzed with 72% sulfuric acid for 1 h, they were diluted to 4% and placed into an autoclave for 1 h at 121 °C. The content of monosaccharides in the liquid was analyzed by high performance liquid chromatography (HPLC). 

The ultraviolet–visible (UV-Vis) spectrophotometer was used to determine the acid-soluble lignin content, and the solid residue was acid-insoluble lignin. For HPLC detection, an Aminex HPX87-H column (Bio-Rad, Hercules, CA, USA) was used. The column temperature was 65 °C, the mobile phase was 0.005 mol/L H_2_SO_4_, and the flow rate was 0.6 mL/min. A differential refractive index detector was used. A determination of the acid-soluble lignin was carried out using the UV-Vis spectrophotometer (UV-Vis, T6 series, Beijing spectrum analysis General Instrument Co., Ltd., Beijing, China) at 205 nm. Each sample was measured twice and the results are averaged.

Detection of reducing sugar in the pre-hydrolyzed solution [23]. 3,5-Dinitrosalicylic acid (DNS) was used as a color indicator, and the content was analyzed at 520 nm by the same T6 UV-Vis spectrophotometer.

#### 2.5.2. FTIR Spectroscopy

The samples were freeze-dried and ground into powder. The KBr and samples were mixed and pressed into thin sheets. Finally, the sheets were scanned by Fourier Transform infrared spectroscopy (FTIR, Nicolet iS10, Thermo Fisher Scientific, Waltham, MA, USA) in the frequency range of 4000–400 cm^−1^ with a resolution of 4 cm^−1^.

#### 2.5.3. Scanning Electron Microscope (SEM) Analysis

Eucalyptus chips with different pretreatments were dried by liquid nitrogen freeze-drying. After this, they were magnified 2000 times with a Scanning Electron Microscope (SEM) (HITACHI 3400-I, Tokyo, Japan) to observe the surface morphology of the dried wood chips after pretreatments.

#### 2.5.4. X-ray Diffraction Method (XRD)

The dried samples were ground into power. Then the crystallinity of samples was analyzed by X-ray diffractometer (XRD) (D8-FOUS, Bruker, Kalsruhe, Germany). The scattering angle (2θ) ranged from 10° to 50° with 0.02 increments. The crystallinity index (CrI) was calculated as the following Equation (1) [24]:CrI = (I_002_ − I_am_)/I_002_ × 100%(1)
where I_002_ is the intensity of the crystalline peak at about 2θ = 22°–23°, attributed to crystalline regions and I_am_ is the intensity at about 2θ = 18°–19°, attributed to amorphous regions.

#### 2.5.5. Fiber Quality Analysis

A small number of samples were mixed with 20 mL of deionized water, and then the mixture was shaken in the test tube vigorously until there are no bundled fibers. Lastly, the fine fiber content, fiber length and fiber width of the sample was analyzed with a fiber quality analyzer (OpTest Equipment Inc., Hawkesbury, ON, Canada).

#### 2.5.6. Fock Reactivity and Degree of Polymerization

Fock reactivity was determined similarly with a previous report [25]. Simply speaking, 0.500 (± 0.004) g of the dried pulp sample was dispersed in 50 mL of 9% NaOH and 1.3 mL of CS_2_. The mixture was shaken at 19 °C under 250 rpm for 3 h. Then, deionized water was added to make the solution a total weight of 100 g. After centrifugation at 5000 rpm for 15 min, 10 mL of supernatant was added into 3 mL of 20% sulfuric acid to neutral. The regenerated cellulose was obtained after 20 h. Then, 20 mL 68% sulfuric acid was added to the solution with stirring for 1 h to acidify the cellulose. Then, 10 mL K_2_Cr_2_O_7_ (1/6 mol/L) was added into the solution, and refluxed for 1 h to oxidize the cellulose. Finally, excess KI solution was added. The final solution was titrated with sodium thiosulfate with starch as an indicator. The calculation method is as Equation (2):(2)$$\begin{array}{c} (\mathrm{volume}\;\mathrm{of}\;K_{2}{\mathrm{Cr}}_{2}O_{7}\times \mathrm{concentration}\;\mathrm{of}\;K_{2}{\mathrm{Cr}}_{2}O_{7}-\;\mathrm{volume}\;\mathrm{of}\;\mathrm{Na}2S2O3\times \\ \mathrm{concentration}\;\mathrm{of}\;\mathrm{Na}2S2O3\times \frac{1}{6})\times 162\times \frac{1}{4}\times \frac{\frac{100}{10.4}}{\mathrm{weight}\;\mathrm{of}\;\mathrm{pulp}\;\mathrm{sample}}\;\times \;100\% \end{array}$$

The polymerization degree of the dissolving pulps were calculated from the viscosity, PD^0.905^ = 0.75 [η]. Each sample was conducted twice.

#### 2.5.7. Other Test Analysis

All other analyses were based on the Chinese national standards. Pentosan measurement was based upon GB/T 745-2003. Specifically, a 1 g sample was azeotroped with a certain amount of 12% hydrochloric acid (distillated 30 mL every 10 min, 300 mL in total), to convert pentosan to furfural. Then the furfural was interacted with Br_2_, and the excess Br_2_ was reduced with KI. Lastly, the reduced I_2_ was titrated with Na_2_S_2_O_3_ (0.1 mol/L). The conversion of furfural to pentosan was multiplied by a factor of 1.88. One molecule of furfural consumed 4.05 molecules of Br_2_. As Equation (3):(3)$$\begin{array}{c} \mathrm{Pentosan}\;\mathrm{content}\;(\%)\;=\;1.88\times \mathrm{volume}\;\mathrm{of}\;\mathrm{Na}2S2O3\times \mathrm{concentration}\;\mathrm{of}\;\mathrm{Na}2S2O3\times \\ \frac{0.024}{\mathrm{weight}\;\mathrm{of}\;\mathrm{sample}} \end{array}$$

The Kappa number was determined by the method in Appendix A of the GB/T 1546-2004 [3]. Specifically, the sample of pulp was oxidized with 0.02 mol/L acid potassium permanganate standard solution for 5 min at 25 °C, and excess KMNO_4_ was reduced with KI. Lastly, the reduced I_2_ was titrated with Na_2_S_2_O_3_ (0.1 mol/L). As Equations (4) and (5):(4)$$\mathrm{Kappa}\;\mathrm{number}\;=\;\frac{\mathrm{volume}\;\mathrm{of}\;\mathrm{Na}2S2O3\times \mathrm{concentration}\;\mathrm{of}\;\mathrm{Na}2S2O3}{0.02\times 5}\times \frac{ʄ}{\mathrm{weight}\;\mathrm{of}\;\mathrm{pulp}\;\mathrm{sample}}$$
(5)$$\mathrm{Log}\;(ʄ)\;=\;0.00093\times \left(\frac{\mathrm{volume}\;\mathrm{of}\;\mathrm{Na}2S2O3\times \mathrm{concentration}\;\mathrm{of}\;\mathrm{Na}2S2O3}{0.02\times 5}-50\right)$$

The viscosity of the samples were measured using cupriethylenediamine (CED) solution as the solvent [26]. Specifically, 0.5 mg of pulp was dissolved in 25 mL of CED and 25 mL of water. After this, the time that the solution flowed out of the capillary viscometer was recorded with a stopwatch. As Equations (6) and (7):(6)Relative viscosity = viscometer constant×time
(7)$$\mathrm{Viscosity}\;=\;\frac{\left[\eta \right]\;\times \;\rho }{\mathrm{Mass}\;\mathrm{concentration}\;\mathrm{of}\;\mathrm{pulp}\;\mathrm{in}\;\mathrm{solution}}$$

The relationship of [η] × ρ and relative viscosity was referred for the table A [26].

The α-cellulose measured was based on GB/T 744-1989 [27]. Specifically, the mercerization treatment of the 2 g sample of pulp was conducted by 17.5% NaOH (*w*/*w*) and dissolved by 9.5% NaOH (*w*/*w*), and the residue was α-cellulose.

Whiteness was tested according to GB/T 8940.2-1988 [28], and two handsheets were prepared for whiteness testing, and the whiteness test was conducted with a whiteness meter (WS-SD, WENZHOU INSTRUMENTS & METERS CO., LTD, Wenzhou, China). Each sample was conducted twice. 

Each test was conducted twice and the final results were taken by the average of the measurements.

### 2.6. Evaluated Treatments of Dissolving Pulp Production of Acetic Acid Pretreatment

The recovery rate of reducing sugar was calculated with reducing sugar concentration multiplying by the volume of recovered liquid. The recovery method of acetic acid lignin were by adding deionized water (5:1) [29]. The lignin sulfate was obtained by adjusting the pH of the waste liquid to 2, followed by washing and centrifugation [30].

## 3. Results

### 3.1. Pre-Experimental Analysis

In the process of pretreatment and hydrolysis, the degradation of lignocellulose is very complicated. Some researchers use the reaction time and the function of temperature and pH as the combined severity (CS) for the evaluation of dilute sulfuric acid pretreatment [31,32,33]. The experiment aimed to explore the effect of different acid treatments on wood fibers, so it is necessary to maintain consistency of other factors. The purpose of the design pre-experiment is to determine the optimal experimental conditions without the need to design a complex experiment such as CS. Table 2 shows the change of pentosan content and solid yield during the pretreatment at different temperatures and pH.

With the temperature increased from 150 °C to 160 °C at the pH of 2, the solid pentosan content and solid yield decreased by 0.70–1.66%, and 0.18–2.50%, respectively. When the temperature rises from 160 °C to 170 °C, the pentosan content decreases by 0.75–1.12%, and the yield decreases by 1.42–3.08%. Compared the temperature from 150 °C to 160 °C with the temperature from 160 °C to 170 °C, the pentosan percentage is reduced more (up to 0.54% more), and the yield is reduced less (up to only 0.58% more). The optimized temperature was 160 °C. At the same temperature (160 °C), when the pH increased from 2 to 3, the pentosan increased by 0.34–2.76%, and the yield changed from 0.99–2.78%. Thus the optimized condition (pH = 2 and 160 °C) demonstrated a lower pentosan content and a higher yield.

### 3.2. Pretreatment

#### 3.2.1. Analysis of Solid and Hydrolysate after Different Pretreatment

Table 3 showed the main components in the solid residue and hydrolysate before and after pretreatment. The pretreatment can reduce the hemicellulose content in the lignocellulose fiber, and the addition of the acid is more effective than hot water pretreatment to remove the xylan, which is consistent with other researchers’ conclusions [9]. When the pH value is the same, the removal effect of organic acid on xylan is better than that of sulfuric acid, and after pretreatment with different acids, the glucan content in solids with organic acid treatment is higher than that of sulfuric acid. Various studies reported that the dicarboxylic acid properties of oxalic acid and maleic acid have excellent selectivity for the hydrolysis of hemicellulose [34,35]. Among the organic acids, acetate is the worst to remove xylan, but the advantage of acetic acid pretreatment can dissolve more lignin than hot water pretreatment and other acid pretreatment. It has been reported that acetic acid can act as a solvent for lignin during the treatment to form acetic acid lignin [36]. The mechanism is that CH_3_CH_2_OO– enters the lignin macromolecular network at high temperature, and is inserted into nodes such as –O–and –C_1⁄4_O. This reaction causes the joint to loosen and destroy the macromolecular network, so the lignin is dissolved in the acetic acid solution [37]. 

The pH of the hot water pretreated hydrolysate after the reaction decreased to 3.27. This is because of the fact that, under high temperature and high pressure conditions, the acetyl group of hemicellulose is destroyed, and acetic acid and other organic acids are produced, which decreases the pH of the system [38]. After acid pretreatment, the pH change may be caused by three reasons: (1) hemicellulose autocatalytic hydrolysis reaction produces organic acid to lower the pH; (2) hemicellulose acid hydrolysis causes H^+^ concentration to decrease; (3) metal cations and anions of organic weak acids in the hydrolysate constitute a buffer system [39]. After the eucalyptus was pretreated with acid, the pH of the hydrolysate increased slightly. This might be due to the consumption of H^+^ during the hemicellulose hydrolysis. In addition, the smaller the hydrolysis equilibrium constant Ka of the weak acid used for pretreatment, the smaller the pH change of the hydrolysate. However, the pH value change after the sulfuric acid pretreatment is not the largest. It may be because the metal ions in the raw material will consume the weak acid and reduce the acid concentration, but the sulfuric acid will not form a buffer system with the metal ions. Also, acid pretreatment releases more reducing sugars in the liquid component, because the addition of acid into the pretreatment increases the dissolution of xylan in the liquid [40,41,42]. Also, the acetic acid pretreatment process recovered less reducing sugars content in the hydrolysate than any other acid pretreatment process, which may be because acetic acid is not effective to remove xylan than other organic acids, resulting in a lower xylose content in the acetic acid treated hydrolysate [28]. Other organic acids pretreatment can produce more reducing sugars than sulfuric acid pretreatment. According to Lu’s research, the activation energy of organic acid degradation is higher, and the degradation of glucose and xylose in the hydrolysate with organic acid is less than that of sulfuric acid [21].

#### 3.2.2. FT-IR Analysis

Figure 1 shows the FTIR spectra of pretreated eucalyptus and raw eucalyptus chips. Stretching vibration in the spectra could be assigned to different groups of lignocellulose components: 1594, 1504 and 1273 cm^−1^ to lignin, 1421, 1319, 1229 and 1106 cm^−1^ to lignin and carbohydrates, and 3350, 2898, 1735, 1370, 1056 1026, and 892 cm^−1^ to carbohydrates [24,28,43,44]. The peaks at 1735 and 1229 cm^−1^ are related to the C=O stretching vibration and C–O stretching vibration in hemicellulose. The decreased peak intensity demonstrated the effective removal of hemicellulose during pretreatment. A slight increase at 1421 and 892 cm^−1^ bands which represents –CH_2_ shear vibration and β-D-glucoside in cellulose, respectively, indicating a slight increase of the crystallinity of cellulose, and additionally a slight increase of crystalline (1370 cm^−1^) and amorphous (2898 cm^−1^) regions of cellulose. The C–O bending vibration at 1273 cm^−1^ in lignin and the slight increase of the characteristic band of aromatic ring at 1594 and 1504 cm^−1^ suggested that part of lignin was released and coprecipitated [44]. The lignin precipitation effect was especially obvious than other samples.

#### 3.2.3. SEM Analysis

From Figure 2, it is clear that there are many spherical droplets on the fiber surface in the SEM image. There are reports that this kind of droplet is due to the higher temperature which will dissolve lignin from the fibers in the pretreatment stage, and will form lignin droplets when they meet the water and reattach to the surface [45]. The hot water pretreatment method and acetic acid pretreatment have a lot of lignin attached to the fiber surface. And the lignin droplets on AAP fibers form a film on the fiber surface. The reason of AAP may be the combination of solvent and the acidity of acetic acid. In the pretreatment stage, the acidity of acetic acid degraded the hemicellulose in the fiber, which increased the fiber pores and accelerated the lignin dissolution [46,47,48]. While the fiber surface of CAP, MAP and OAP showed less lignin droplets than AAP, the disruption on the fiber surface could be clearly observed.

#### 3.2.4. X-ray Diffractometer Analysis

The crystallinity index obtained by X-ray diffraction showing in Figure 3. It is obviously concluded that the crystallinity index of eucalyptus increased after pretreatment (crystallinity index increased by 10–14.2%). This means pretreating could change the lignocellulose structure considerably [49]. The CrI may be indirectly showing the removal of hemicellulose [50], such as in the samples of OAP, MAP and CAP, which have a higher crystallinity index corresponding to lower xylan. However, the crystallinity index of AAP is the smallest, even smaller than HP, which has the largest xylan content. The reduced crystallinity of biomass may be caused by the amorphous acetic acid lignin heavily dissolved and attached to the fiber surface after acid pretreatment. 

### 3.3. Cooking

#### 3.3.1. Kappa Number and Residual Alkali

Table 4 shows the Kappa number, the amount of residual alkali in the black liquor, and the yield from the raw materials to the cooking chemical pulp after different pretreatments under the same conditions. From Table 4, the Kappa number of HP-CP is significantly higher than that of cooking pulp after acid pretreatment. The kappa number of the chemical pulp after acid pretreatment is low, which indicates that lignin can be removed well under this cooking condition, and there is still more residual alkali in the black liquor, which indicates that the cooking liquor is basically excessive. Except for HP-CP and AAP-CP, the yields of other pulps were lower, indicating that the hydrothermal method and acetic acid pretreatment had less degradation to the cellulose.

#### 3.3.2. Fiber Length Analysis of Cooking Pulp

Table 5 shows the fiber morphology of cooking pulp after different pretreatment. HP-CP has the smallest content of fine fibers, the longest average length, and the widest average width, indicating that this pretreatment method has the least damage to the fibers. Except for AAP-CP, Other organic acid pretreatment cooking pulp, inferior fiber properties (short, thin and more fine fibers) was obtained compared with water and sulfuric acid pretreatment cooking pulp. This shows that in addition to acetic acid, other organic acids have a greater degree of fiber breakage. The fiber quality in AAP-CP is the best among all acid-pretreated cooking pulps, but the Kappa number is not the largest, indicating that the content of lignin in AAP-CP is small, and the fiber damage is small. It is speculated that the lignin film had attached to the fiber surface after acetic acid pretreatment, which occurrence can protect the fiber and reduce the damage of the cooking liquid.

### 3.4. Bleaching

#### 3.4.1. Fiber Length Analysis of Bleaching Pulp

Table 6 shows the fiber quality after bleaching. It can be concluded that the bleached fine fiber content of the chemical pulp increases, and the fiber length and width become smaller, indicating that the bleaching process causes great damage to the fibers. HP-BP has the smallest amount of fine fibers, the longest fiber length, and the widest width. Similarly, with the exception of AAP-BP, other organic acids pretreatment dissolving pulp have more fine fibers, shorter fiber lengths and narrower widths than sulfuric acid pretreatment-dissolving pulps. AAP-BP exhibits best fiber properties in the organic acid pretreatment-dissolving pulp. The effect of the pretreatment stage on the degree of fiber damage can continue into the bleaching stage.

#### 3.4.2. Fock Reactivity and Degree of Polymerization of Dissolving Pulp

The reaction performance of dissolving pulp can be understood as the ability of pulp to react with all reactants. It mainly depends upon the accessibility of cellulose [51]. As can be seen in Figure 4, except for AAP-BP, the Fock reactivity of the dissolving pulp obtained by pretreatment with other organic acids can reach more than 90%. The HP-BP Fock reactivity is only 60.97%, which is much lower than other dissolving pulps. MAP-BP with the largest Fock reactivity has the smallest polymerization degree, and AAP-BP with the smallest Fock reactivity has the largest polymerization degree.

As can be seen from Table 7, the degree of polymerization and the Fock reactivity show opposite changes, similar to the results of Liu [52]. Fock reactivity and the degree of polymerization seem to be related to the fine fiber content and fiber length in the fiber. HP-BP with the smallest fine fiber content, the longest fiber length, and the widest width, also has the smallest Fock reactivity. The average length of SAP-BP and AAP-BP fibers is similar, but the fine fiber content of SAP-BP is higher than AAP-BP. At the same time, the higher Fock reactivity indicates the same rule comparing CAP-BP and OAP-BP. However, comparing OAP-BP and OAP-BP, although OAP-BP has a smaller content of fine fibers, the average length of MAP-BP fibers is shorter, while MAP-BP has a slightly higher Fock reactivity. It seems that the fiber length has a greater effect on the Fock reactivity of the dissolving pulp than the fine fiber content. But, fibers with a large fiber content and short fiber length have a high Fock reactivity and a low degree of polymerization.

#### 3.4.3. Dissolved Pulp Index

Table 7 shows the basic parameters of the dissolving pulp prepared under different pretreatment conditions. The AAP-BP has less pentosan content (2.14%), higher α-cellulose content (98.28%), moderate viscosity (396.9 mL/g), higher whiteness (88.68% ISO) and higher total yield (31.66%), which makes it the best quality of dissolving pulps. Although the MAP-BP has a higher Fock reactivity (98.87%), it becomes an unqualified dissolving pulp because of low α-cellulose (83.58%) and low viscosity (301.5 mL/g). The CAP-BP and OAP-BP have lower pentosan content (2.23% and 1.96%), higher α-cellulose (92.75% and 83.58%) and higher Fock reactivity (98.87% and 96.20%), but the total yields are too low to be applied in actual production (25.85% and 22.29% respectively). The HP-BP and SAP-BP dissolving pulp index can only be regarded as qualified. By the way, all dissolving pulps have high whiteness (>87% ISO).

#### 3.4.4. Extensive Utilization of Eucalyptus during AAP-Dissolving Pulp Production

It can be concluded that acetic acid pretreatment exhibited the best pretreatment results. In order to attain the extensive utilization of biomass, the mass balance of the acetic acid pretreatment dissolving process was evaluated. The hydrolysate could recover reducing sugar and acetic acid lignin (acetic acid lignin: pretreated liquid = 0.017 g:50 mL). The solid is used to produce high quality dissolving pulp. The black liquor produced during the cooking process can recover the sulfate lignin by acid precipitation (sulfate lignin: black liquor = 2.743 g:50 mL). The approximate flowchart is shown in Figure 5. 43.8% polysaccharide (12.14% reducing sugar and 31.66% dissolving pulp) and 22.24% lignin (0.29% acetic acid lignin and 21.95% sulfate lignin) was recovered. Biomass utilization could reach up to 66.04%.

## 4. Conclusions

In this paper, the effect of various organic acid pretreatment for Kraft dissolving pulp production was comparatively investigated. The pretreatment condition for organic acid pretreatment was optimized (160 °C, 2 h, pH = 2). Compared with other organic acids (citrate acid, maleic acid and oxalic acid), acetic acid alleviates cellulose degradation due to the solvent effect. Besides, acetic acid exhibited better hemicellulose removal over autohydrolysis pretreatment. For the Kraft dissolving pulp properties, acetic acid pretreatment demonstrated higher yield (31.66%) and higher α-cellulose (98.28%) over other organic acid pretreatment, and lower hemicellulose content (2.14%) compared with autohydrolysis pretreatment. Also, the Fock reactivity attained 81.03%. Acetic acid pretreatment-derived dissolving pulp can meet the commercial index of dissolving pulp. The dissolving pulp obtained by other organic acid pretreatment had more fine fiber content due to the excessive acid hydrolysis of cellulose. Moreover, the extensive utilization of biomass was evaluated. 43.8% polysaccharide (12.14% reducing sugar and 31.66% dissolving pulp) and 22.24% lignin (0.29% acetic acid lignin and 21.95% sulfate lignin) were recovered. Biomass utilization could reach 66.04%. Acetic acid pretreatment process fulfilled the biorefinery concept and showed the potential application for the Kraft dissolving pulp. Overall, the acetic acid process is a promising pretreatment method for extensive biomass utilization. 

## Figures and Tables

**Figure 1 materials-13-00361-f001:**
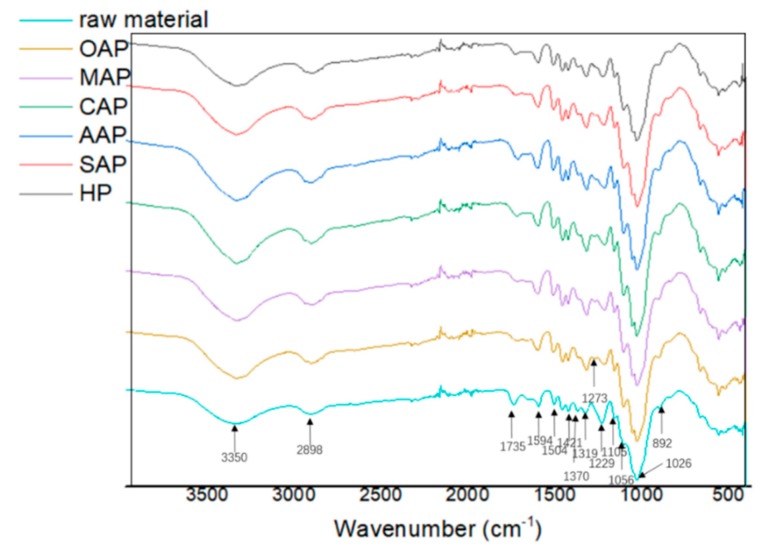
The Fourier transform infrared spectroscopy (FTIR) spectra of raw material and pretreated eucalyptus.

**Figure 2 materials-13-00361-f002:**
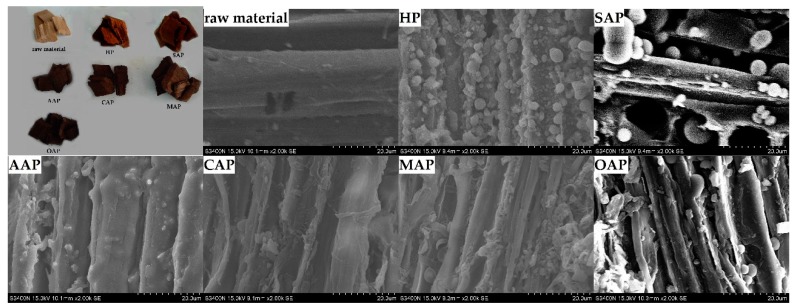
Eucalyptus chips of the scanning electron microscopy (SEM) image after different pretreatment.

**Figure 3 materials-13-00361-f003:**
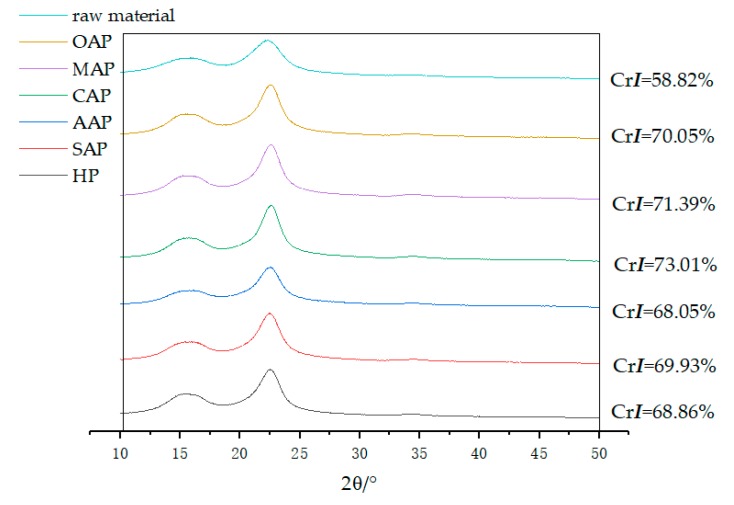
X-ray diffractometry (XRD) plots for raw material and pretreated eucalyptus.

**Figure 4 materials-13-00361-f004:**
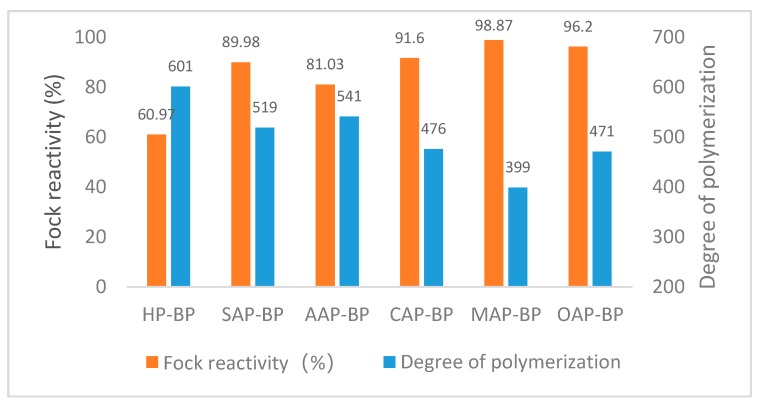
Fock reactivity and degree of polymerization of different dissolving pulps. BP means dissolving pulp after bleaching, SAP means sulfuric acid pretreatment, AAP indicates acetic acid pretreatment, while CAP represents citric acid pretreatment, MAP means maleic acid pretreatment and OAP implies oxalic acid pretreatment.

**Figure 5 materials-13-00361-f005:**
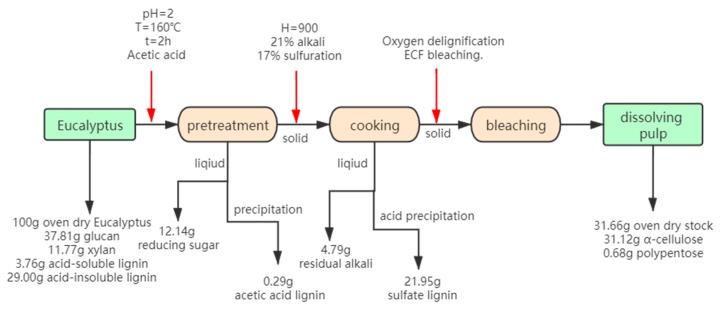
Flow chart of full utilization of biomass after pre-acetic acid.

**Table 1 materials-13-00361-t001:** Oxygen delignification and elemental chlorine free (ECF) bleaching process parameters.

Stage	Stock Conc. (%)	Reagent Type	Temp (°C)	Time (h)	pH
O	12	O_2_ (0.5 MPa)	100	1	10–11
		NaOH (3%)			
		MgSO_4_ (0.05%)			
D	10	ClO_2_ (0.9%)	70	2	3.5–4.5
E	10	NaOH (0.3%)	70	1	10.5–11.5
P	10	H_2_O_2_ (0.3%)	70	1	10–11
		MgSO_4_ (0.05%)			
		Na_2_SiO_4_ (3%)			

**Table 2 materials-13-00361-t002:** Pre-experimental results.

Sample ^1^	pH = 3		pH = 2					
	160 °C		150 °C		160 °C		170 °C	
	Pentosan Content	Yield	Pentosan Content	Yield	Pentosan Content	Yield	Pentosan Content	Yield
	(%)	(%)	(%)	(%)	(%)	(%)	(%)	(%)
P-SAP	7.39	76.87	8.17	77.00	6.52	74.50	5.39	72.47
P-AAP	6.24	73.99	7.33	73.98	5.9	71.73	4.94	69.11
P-CAP	7.64	75.34	5.98	74.22	4.89	72.68	4.13	69.73
P-MAP	7.58	75.32	6.71	76.32	5.45	72.55	4.64	71.13
P-OAP	6.21	75.79	6.35	74.98	5.65	74.80	4.66	71.72

^1^ P- means pre-experiment’-SAP means sulfuric acid pretreatment, -AAP means acetic acid pretreatment, -CAP means citric acid pretreatment, -MAP means maleic acid pretreatment, -OAP means oxalic acid pretreatment.

**Table 3 materials-13-00361-t003:** Raw material composition and pretreatment results of different methods.

Sample ^1^	Solid Content ^2^				Hydrolyzate	
	Glucan	Xylan	Acid-Soluble Lignin	Acid-Insoluble Lignin	pH	Reducing Sugar
	(%)	(%)	(%)	(%)		(g/L)
raw material	37.81	11.77	3.76	26.40	-	-
HP	37.86	3.91	2.05	26.49	3.27	7.82
SAP	36.12	3.11	1.99	25.98	2.31	18.13
AAP	38.09	2.48	1.59	23.36	2.24	14.59
CAP	37.55	1.99	1.71	25.41	2.27	19.62
MAP	36.35	2.44	1.86	26.31	2.47	18.56
OAP	37.47	1.68	1.86	26.01	2.65	20.18

^1^ HP means hot water pretreatment; SAP means sulfuric acid pretreatment, AAP means acetic acid pretreatment, CAP means citric acid pretreatment, MAP means maleic acid pretreatment, OAP means oxalic acid pretreatment. ^2^ The solid content is based on raw materials.

**Table 4 materials-13-00361-t004:** Cooking results.

Sample ^1^	Kappa Number	Residual Alkali Content (g/L)	Yield ^2^ (%)
HP-CP	13.6	14.00	39.54
SAP-CP	6.2	12.55	31.37
AAP-CP	7.6	12.04	33.28
CAP-CP	7.1	11.50	26.91
MAP-CP	6.5	12.57	30.29
OAP-CP	8.3	12.32	23.51

^1^ CP means that the pretreated eucalyptus is cooked to obtain chemical pulp, SAP means sulfuric acid pretreatment, while AAP signifies acetic acid pretreatment, CAP means citric acid pretreatment, MAP indicates maleic acid pretreatment and OAP denotes oxalic acid pretreatment. ^2^ Yield refers to the yield of raw materials after pretreatment and cooking.

**Table 5 materials-13-00361-t005:** Chemical pulp fiber quality analysis.

Sample ^1^	Content of Fine Fibers (%)	Average Length (mm)	Average Width (μm)
HP-CP	13.06	0.612	16.1
SAP-CP	18.27	0.557	15.9
AAP-CP	17.78	0.575	15.7
CAP-CP	24.13	0.536	15.3
MAP-CP	21.51	0.526	15.5
OAP-CP	23.41	0.532	15.3

^1^ CP means that the pretreated eucalyptus is cooked to obtain chemical pulp, SAP means sulfuric acid pretreatment, while AAP signifies acetic acid pretreatment, CAP means citric acid pretreatment, MAP indicates maleic acid pretreatment and OAP denotes oxalic acid pretreatment.

**Table 6 materials-13-00361-t006:** Dissolved pulp fiber quality analysis.

Sample ^1^	Content of Fine Fibers (%)	Average Length (mm)	Average Width (μm)
HP-BP	15.61	0.582	15.3
SAP-BP	24.41	0.517	14.6
AAP-BP	23.69	0.512	14.9
CAP-BP	25.77	0.493	14.6
MAP-BP	29.92	0.479	14.5
OAP-BP	31.32	0.495	14.3

^1^ BP indicates dissolving pulp after bleaching, SAP means sulfuric acid pretreatment, while AAP denotes acetic acid pretreatment, CAP means citric acid pretreatment, MAP represents maleic acid pretreatment and OAP infers oxalic acid pretreatment.

**Table 7 materials-13-00361-t007:** Technical specifications of different dissolving pulp.

Sample ^1^	Polypentose	α-Cellulose	Viscosity	Whiteness	Yield ^2^	Total Yield ^3^
	(%)	(%)	(mL/g)	(%ISO)	(%)	(%)
HP-BP	3.31	89.81	436.2	87.99	97.70	38.59
SAP-BP	2.62	92.22	382.2	88.32	94.76	29.76
AAP-BP	2.14	98.28	396.9	88.68	95.06	31.66
CAP-BP	2.23	92.75	353.5	87.02	96.09	25.85
MAP-BP	1.96	83.58	301.5	87.27	94.09	28.51
OAP-BP	1.99	96.17	349.9	87.00	94.86	22.29

^1^ BP means dissolving pulp after bleaching; SAP indicates sulfuric acid pretreatment, while AAP designates acetic acid pretreatment, and CAP- means citric acid pretreatment; MAP means maleic acid pretreatment, OAP refers to oxalic acid pretreatment. ^2^ Yield indicates only the yield in the bleaching stage. ^3^ Total yield indicates the yield of wood chip raw materials after pretreatment, cooking and bleaching.

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
