# Peer review of "Comparative Evaluation of Organic Acid Pretreatment of Eucalyptus for Kraft Dissolving Pulp Production"

_materials, 2020, doi:10.3390/ma13020361_

Round 1

Reviewer 1 Report

In this work acid pretreatment of lignocellulose by several common organic acids and sulfuric acid was performed. Besides, the hydrothermal pretreatment was applied.  It was confirmed that under acidic conditions, hemicellulose can be removed and sugars can be recovered.  However, it is known from decades.  

The obtained results are of very low scientific novelty as the main techniques to pretreat lignocellulose  - physical (e.g.,  hydrothermolysis,  microwaves, ultrasound), physicochemical (e.g., steam explosion, biological (fungi, bacteria, microbes) and chemical (e.g., acid, alkali, solvents, ozone) are very well known and broadly described in the literature, e.g. Bioresour Technol, 199 (2016), pp. 113-120;  Fuel Process Technol, 160 (2017), pp. 196-206 or Prog Energy Combust Sci, 42 (2014), pp. 35-53.  

In the reviewed work authors should expand the Introduction to cover the recent adcvances in the field, as well perform critical discussion of the literature results. The Discussion part definitely needs to be expanded, including the discussion  on the structure-property relationships. XRD analysis should be performed to discuss the crystallinity features.  Conclusion has to be re-written as it is very poor and resembles a short abstract, which is not the conclusion. 

English needs major revison, e.g. "Acid treatment of biomass is a more common treatment.",  "Eucalyptus from HuNan YuNong Paper Co., Ltd.." (Section 2.1. - is this a sentence?) or the title "Explore …". 

Author Response

Thanks for your sincere suggestion.

The discussion of main techniques to pretreat lignocellulose is made again in the second paragraph of the introduction. The crystallinity features are supplemented and analyzed (section 3.2.4). The conclusion is rewrited.

I am very sorry for my poor English. If there were still any errors, please criticize or correct. Thanks.

Reviewer 2 Report

This manuscript presents a comparison of the pretreatment methods to process eucalyptus biomass. The authors compare the use of several acidic pretreatments. The manuscript is not very accessible to someone who is not familiar with biomass processing. The authors need to add details to the methods section to allow other researchers to repeat their work. Addressing these issues and those that I detail below would improve the quality of the manuscript.

Comments

This manuscript assumes the reader has a significant degree of familiarity with biomass processing. Adding content to the introduction to describe biomass, the use for lignocellulose, and its basic processing briefly would help a more general reader gain a better understanding of the research. Also, the target yields for the different biomass components should be described.

The SEM analysis should include an image for an untreated control with no pretreatment and images for each sample group.

What is the concentration of the respective acids used? Only a pH value is stated, which does not describe concentration. There should be descriptions of each solution in the methods section.

More details are required in methods section about techniques performed and equipment used. Sections 2.4.2 – 2.4.5 need more details. The evaluated treatments are not mentioned in the methods section. These should be clearly described.

How many samples were analyzed per condition? This is not stated.

The discussion section is very brief and there is content in the results section that should be moved to discussion. The content in the results section can be better organized to make it easier for the reader to follow.

Statistical analysis should be used to evaluate the results.

Minor Comments

P5 lines 157-158 have sentence fragment and/or run on sentence.

The y-axes in Figure 2 should be labeled.

The results section contains text from the template at the beginning.

The title needs to be revised to improve grammar. Making the first word exploring could solve the problem.

Author Response

Thanks for your comments. I'm sorry for my simple experiment procedure in the manuscript. I have supplemented the experimental details in the manuscript.

Thanks for pointing out the rigor of my experiment. The SME image of raw material and other sample have been added (section 3.2.3). The pH values were measured by the pH meter at room temperature. The solution were diluted with concentrated acid and water. The specific concentration could only be calculated. In the part of experimental details,I have supplemented, including techniques performed, equipment used and the method of evaluated treatment( in the sections 2.4). For some experment data, I have conducted twice and the result were averaged. The content in the results section have been reorganized. The data in Figure 4 were based on eucalyptus raw materials, but I use statistical analysis for the analysis of results.

For minor comments, I have modified. Thanks for your comments, if there were any errors, please criticize again. (I am very sorry for my poor English.)

Reviewer 3 Report

The aim of the work is to find the different ways of treatment of lignocellulose by organic acids, sulfuric acid and hydrothermal pretreatment. The main contributions are that under acidic conditions, hemicellulose can be effectively removed and more reducing sugars can be recovered, as well as that acetic acid and other organic acids are more selective than sulfuric acid to remove xylan and recover more reducing sugars. However the disadvantage is that the latter cause damage to the fiber. Also, acetic acid cannot remove much xylan and recover many reducing sugars like other organic acids, but it can be used as a solvent to dissolve more lignin and form a film on the fiber surface. Some other new results comparing different acids and treatment ways are also presented and discussed.

Areas of strength: Research is appropriately designed in terms of both research idea and its implementation. The work has considerable applied impact. Suitable investigation tools are used. The work is well arranged, clearly written, accurately typeset, tables and figures are informative and clear, literature list is appropriate and up-to-dated.

Areas of weakness: To strengthen the work and results obtained, the authors could add the techniques used, for instance, by infrared measurements (FTIR), the technique informative to the processes under investigation. These could add the structural information obtained by SEM, by chemical information at the molecular level.

The work can be published in Materials in its present form.

Author Response

Thanks for your approval. SEM image have been supplemented. And the analysis of FTIR have been added.

There are still many shortcomings in my work, please continue to criticize and correct

Round 2

Reviewer 1 Report

The revised version of the manuscript addressed the reviewer's comments on experimental section, however, Introduction part should still be expanded to cover the recent advances in the field, as well to perform critical discussion of the literature results.

English is poor - the manuscript should be thoroughly checked by a native speaker, and the certificate provided. 

Author Response

Thanks for your comments

We have added more references about recent progress related to pretreatment in the Introduction part. Also, the advantages and disadvantages are compared. We have marked the revised part in red.

We have checked and revised the language carefully, and all the revised part are labelled in green.

Reviewer 2 Report

The authors have improved the quality of this revised manuscript. There are still some more details that should be addressed to improve the quality of the manuscript. 

Comments

The authors still need to provide more details for methods. What is brand and model of photometer? What is brand and model of FTIR? These details need to be provided so a reader understands what equipment was used.

For the Chinese standards mentioned, more detail should be provided for the methods used so that someone who cannot directly access the standard has a clear idea of what was performed.

More detail for the FTIR results are needed. What are the molecular vibrations for the stated wave numbers? This would give the reader deeper insight into the effect of the treatments.

The methods section text could be revised. The methods in many places reads more like a procedure then a journal article. I suggest to revise the text to make it like an article.

Minor Comments

The results section still contains text from the template at the beginning. Please delete lines 169-170.

Several acronyms need to be defined, including ECF and DEP.

Author Response

Thanks for your comments

We have revised the manuscript accordingly.

photomete is T6 series and from Beijing spectrum analysis General Instrument Co., Ltd. FTIR is Nicolet iS10. Please see the sections (2.5.1 and 2.5.2) marked in red;

For the Chinese standards mentioned,I have added the details of the procedure. Please see the sections marked in red in sections 2.5.6;

We have added more details for FTIR related to various components in the Eucalyptus. The FTIR results of molecular vibrations are explained in sections 3.2.2 marked in red.

The methods section text have been revised. Please see the sections marked in green in sections 2.2, 2.3 and 2.4.

We have removed the content in lines 169-170.

ECF are defined as Elemental Chlorine Free bleaching, and D E P in the bleaching process are defined as chlorine dioxide treatment, alkaline extraction and hydrogen peroxide, respectively. Please see the sections marked in green in sections 2.4.